# Ultrasmall ATP-Coated Gold Nanoparticles Specifically Bind to Non-Hybridized Regions in DNA

**DOI:** 10.3390/nano13243080

**Published:** 2023-12-05

**Authors:** Liat Katrivas, Asaf Ben-Menachem, Saloni Gupta, Alexander B. Kotlyar

**Affiliations:** Department of Biochemistry and Molecular Biology, George S. Wise Faculty of Life Sciences and Nanotechnology Center, Tel Aviv University, Ramat Aviv, Tel Aviv 69978, Israel; liatkatrivas@mail.tau.ac.il (L.K.); asafb2@mail.tau.ac.il (A.B.-M.); salonigupta@mail.tau.ac.il (S.G.)

**Keywords:** gold nanoparticles, Puc19, DNA–nanoparticle conjugate, AFM, TEM

## Abstract

Here we report the synthesis of ultrasmall (2 nm in diameter) ATP-coated gold nanoparticles, ATP-NPs. ATP-NPs can be enlarged in a predictable manner by the surface-catalyzed reduction of gold ions with ascorbate, yielding uniform gold nanoparticles ranging in size from 2 to 5 nm in diameter. Using atomic force microscopy (AFM), we demonstrate that ATP-NPs can efficiently and selectively bind to a short non-hybridized 5A/5A region (composed of a 5A-nucleotide on each strand of the double helix) inserted into a circular double-stranded plasmid, Puc19. Neither small (1.4 nm in diameter) commercially available nanoparticles nor 5 nm citrate-protected ones are capable of binding to the plasmid. The unique ability to specifically target DNA regions characterized by local structural alterations of the double helix can pave the way for applications of the particles in the detection of genomic DNA regions containing mismatches and mutations that are common for cancer cells.

## 1. Introduction

Studies of spherical gold nanoparticles (GNPs) have been focused on the structure and plasmonic properties of the particles as well as on their technological and biomedical applications (for reviews, see [1,2,3,4,5]). Many nanoparticle-based methods have been developed for the sensitive detection of biomolecules [6,7,8,9] as well as for the delivery of biomolecule-conjugated GNPs into cancer and other pathological cells [5,10,11,12]. The most commonly used GNPs range in diameter from one to one hundred nanometers. The synthesis of GNPs > 5 nm in diameter involves, in most cases, the reduction of gold ions by a weak reductant, citrate, at elevated temperatures or under ambient temperature by strong reductants (borohydride and others). The former approach commonly known as the Turkevich method [13,14], yields particles of about 15 nm in diameter, while the latter yields GNPs of less than 5–6 nm [14,15,16]. Both types of GNPs can be further enlarged by the surface-catalyzed reduction of gold ions with ascorbate (or other mild reductants) [14,15,16]. In this seed-growing reaction the gold ions are being reduced on the particle’s surface, leading to its growth. The above GNPs are commonly protected by citrate, which is characterized by rather low affinity to the gold core and can be rapidly exchanged by higher-affinity ligands: thiols, phosphines, polymers, DNA oligonucleotides, proteins and peptides, and many others. Coating with the above ligands greatly stabilizes the particles, making them resistant to higher (including physiological) salt concentrations in the hundreds of millimolar range [17]. Reducing gold ions in the presence of citrate by either strong or weak reductants does not yield ultrasmall (1–2 nm diameter) gold nanoparticles. The requirement for their formation is the presence of strong (characterized by high affinity to gold) ligands in the synthesis. These ligands stabilize small clusters/nanoparticles formed during the reduction of gold ions, preventing their further growth and aggregation. Two main approaches have been used for synthesis of these ultrasmall nanoparticles. The first, reported by Schmid et al. in 1981 [18], involves the reduction of gold ions by a strong reductant, diborane, in an apolar medium in the presence of triphenylphosphine (PPh_3_). The procedure yielded uniform ultrasmall (1.4 nm in diameter) PPh_3_-capped gold particles composed of 55 gold atoms (Au_55_). The capping ligand can be further exchanged by either hydrophilic phosphines or thiol-containing hydrophilic ligands [19,20,21,22] resulting in water-soluble particles. These particles are commercially available and widely used in chemical and biological research. An alternative approach pioneered by Brust [23] employed thiol-containing ligands, which protect the small clusters/nanoparticles being formed during the synthesis. Uniform particles comprising discrete number of gold atoms and capping thiol ligands have been synthesized and characterized on the atomic level by Kornberg et al. [24,25,26,27,28] rather recently. Capping ligands on the surface of these particles can be replaced by other thiol-comprising compounds [29,30], making it possible to conjugate the particles with thiolated single-stranded (ss) DNA oligonucleotides [25] and other thiol-containing biomolecules. 

Mostly driven by environmental concerns, the use of natural compounds for the preparation of gold nanoparticles (“green particles”) has become popular nowadays (for recent reviews, see [2,3,31,32]). The main advantage of natural compounds over chemical ligands and synthetic polymers is that they are non-toxic, easily degradable, and in most cases inexpensive. ATP, which is known as the energy currency of the cell, was also employed as a capping ligand in the synthesis of GNPs. The reduction of gold ions by borohydride in the presence of ATP was shown to yield relatively uniform gold nanoparticles ranging in diameter from 2.7 to 5 nm [33]. The size of the synthesized particles within the above range was tuned by altering the molar ratio between ATP and Au ions. 

Here we report a method for the synthesis of ultrasmall, highly uniform ATP-coated gold nanoparticles, ATP-NPs. As in the above-cited article [33], the reduction of gold ions by borohydride was carried out in the presence of ATP. The two main differences between our method and the one published earlier [33] are a 10 times lower concentration of gold ions (20 µM) and a higher pH in the synthesis reaction. The synthesis was previously conducted under mild acidic conditions (pH~3.5), while here the synthesis was performed at pH~8.5. Though seemingly small, these alterations enabled us to obtain very uniform ultrasmall (about 2 nm in diameter) GNPs. ATP-NPs do not exhibit a plasmon resonance maximum around 500–520 nm, which is characteristic for bigger nanoparticles including those reported earlier [33]. ATP-NPs can be enlarged in a predictable manner by the surface-catalyzed reduction of gold ions with ascorbate (or another mild reductant) yielding bigger (2.5–5 nm in diameter) uniform particles. ATP-NPs can be efficiently conjugated with short non-modified ss oligonucleotides as well as non-hybridized (unpaired) regions in long (kilobase pairs) double-stranded (ds) DNA. Using AFM, we demonstrated that the incubation of circular ds plasmid comprising a non-hybridized 5A/5A (composed of a 5A-nucleotide on each strand of the double helix) region with ATP-NPs resulted in the binding of one nanoparticle per plasmid. Incubation of the plasmid lacking 5A/5A with ATP-NPs under identical conditions did not lead to formation of the conjugate. Incubation of 5A/5A-containing plasmid with either commercial thiol-coated Au_55_ or 5 nm (in diameter) citrate-protected GNPs also did not result in formation of the conjugate. The unique ability of ATP-NPs to interact with non-hybridized (non-perfectly hybridized) regions in DNA paves the way for applications of the particles in the detection of regions in genomic DNA containing mismatches and characterized by local structural alterations and deformations of the double helix. Since non-perfectly hybridized regions containing multiple mutations and deletions are known to be common characteristics for cancer cells, ATP-NPs can potentially be useful for the selective detection of pathological cells and their eradication by suppressing the nucleic acid repair mechanisms. 

## 2. Materials and Methods

### 2.1. Materials 

Circular 2686 bp DNA pUC19 plasmid was purchased from New England Biolabs (New England Biolabs Ltd., Ipswich, MA, USA). The plasmid was linearized by cleavage with EcoRI and HindIII purchased from New England Biolabs (New England Biolabs Ltd., Ipswich, MA, USA). Oligonucleotides were purchased from Integrated DNA Technologies (IDT Ltd., Shirley, NY, USA). Other reagents and chemicals (unless otherwise stated) were obtained from Sigma-Aldrich (Sigma-Aldrich Ltd., Burlington, MA USA) and were used without further purification.

### 2.2. Synthesis of ATP-NPs

One milliliter of solution containing 2 mM HAuCl_4_ and 0.1 M ATP (filtered through a 3 KDa Amicon-Ultra Centrifugal Filter Unit) was incubated for 15 min under ambient conditions. During the incubation, the color of the solution changed from light yellow to chrome yellow. The mixture was then added to 100 mL of 2 mM KOH (filtered through 3 KDa Amicon-Ultra Centrifugal Filter Units) under constant stirring at high speed followed by the addition of 0.2 mL of freshly prepared 0.1 M NaBH_4_. The light brown solution was left to stir for a further 5–10 min. The synthesized particles were then centrifuged for 10 min at 20 °C in six 15 mL 10 KDa Amicon-Ultra Centrifugal Filter Units at 4000 rpm. The collected filtrant (~1.5 mL; ~0.2–0.3 mL from each filtration unit) containing concentrated particles was diluted into 15 mL of DDW and centrifuged in the filtration unit as described above. The centrifugation–dilution cycle was repeated five times in order to completely remove the unbound ATP. 

### 2.3. Enlargement of ATP-NPs in Solution

All solutions used for the enlargement were filtered through 3 KDa Amicon-Ultra Centrifugal Filter Units. To 3.5 mL of solution containing 2 mM ATP (neutralized by one molar equivalent of KOH) and ATP-NPs synthesized and concentrated as described above (absorbance at 420 nm ≈ 0.17 AU), 80, 160, or 400 µL of 10 mM HAuCl_4_ solution was added during constant stirring followed by the almost immediate (after 1–2 s) addition of 60 µL of 0.1 M L-Ascorbic acid (neutralized with one molar equivalent of KOH). The color of the incubations rapidly (in 10–20 s) changed from brown to red. One minute after the addition of ascorbate, 8 µL of freshly prepared 0.1 M BSPP solution was added to the incubation under constant stirring. The incubation was left to stir for another 10 min and was subsequently concentrated using a 4 mL 10 KDa Amicon-Ultra Centrifugal Filter Unit to a final volume of 150 µL. 

### 2.4. Enlargement of ATP-NPs on Mica Surface

All solutions used for the enlargement were filtered through 3 KDa Amicon-Ultra Centrifugal Filter Units. The conjugate samples were diluted (to an OD of about 5 mAU at 260 nm) with 10 mM Mg-Ac and deposited on a freshly cleaved mica for 5 min. The substrate surface was then rinsed with ~1 mL of ice-cold 20 mM Mg-Ac; a piece of filter paper was used to pull the excess solution out from the mica. Then, 100 µL of a mixture composed of 25 µL of 10 µM HAuCl_4_-1 mM KBr (the components were pre-incubated for 2–3 h at 4 °C), 74 µL DDW, and 1 µL fresh neutral (pH 6–7) solution of 0.1 mL-ascorbate-K was poured on the surface and left for 30 s. The surface was rinsed with 1 mL of ice-cold water and then quickly dried with nitrogen gas. AFM measurements were conducted as described in Materials and methods.

### 2.5. Synthesis of ssDNA–ATP-NP Conjugates 

A 38-base ssDNA, 5′-C*C*C*C*C*CCAAGCAACAGAGGTTGTTAACGTAAGTTATGT-3′, comprising five phosphorothioate nucleotide residues (*) at the 5′ end was modified by N,N′-bis(α-iodoacetyl)-2-2′-dithiobis(ethylamine), BIDBE, as previously described [34,35]. The reaction resulted in the attachment of a disulphide group to the DNA backbone at the phosphorothioate nucleotide position. The disulphide residues were reduced during the 2-h incubation of the modified oligonucleotide with 5 mM dithiothreitol (DTT) under ambient conditions. The oligonucleotide was separated from DTT on a size-exclusion NAP-5 column (Cytiva Inc., Marlborough, MA, USA) equilibrated with 10 mM Citrate-K (pH 6) and 50 mM KCl. The oligonucleotide eluted from the column in the void volume was then incubated with BSPP-treated ATP-NPs in 10 mM Citrate-K (pH 6) and 50 mM KCl for 16 h under ambient conditions. Concentrations of the oligonucleotide and the particles were equal to 10 and 4 µM, respectively. The concentrations were measured spectrophotometrically using extinction coefficients of 369 × 10^3^ M^−1^cm^−1^ and 7.4 × 10^5^ M^−1^cm^−1^ at 260 and 420 nm for the oligonucleotide and the particles, respectively. The conjugates comprising different numbers of DNA strands per particle were separated from one another by gel electrophoresis. Ten minutes prior to loading on the agarose gel, 1 mM BSPP was added to the samples. 

### 2.6. Preparation of pUC(5A/5A) and pUC(5A/5A) Conjugates with ATP-NPs

pUC19 was linearized with EcoRI and HindIII (New England Biolabs Ltd., Ipswich, MA, USA) according to company protocol. The linearized plasmid was purified from the enzymes and other components of the assay on a size-exclusion Sepharose CL-2B (GE HealthCare, Chicago, IL, USA) (5 × 0.9 cm) equilibrated in 50 mM HEPES-K (pH 7.5). The void volume fraction containing the plasmid was collected and ligated with a short dsDNA construct comprising a non-hybridized 5A/5A fragment (Appendix A). The construct was prepared as follows: 5′-P-AATTCAAGCAATTAAGAACAAAAAAACGCAAGGATCGCG-3′ and 5′-P-AGCTCGCGATCCTTGCGTTAAAAAGTTCTTAATTGCTTG-3′ DNA oligonucleotides (the OD of each strand at 260 nm was equal to 80–90 AU) were mixed in equimolar concentrations in 0.1 M LiOH. The mixture (~0.2 mL) was transferred into a dialysis bag and dialyzed against 1 L of 100 mM NaCl, 20 mM Tris-HCl (pH 7.5) for 16 h at ambient temperature. The concentrations of the oligonucleotide construct and the plasmid were measured spectrophotometrically using extinction coefficients of 680 and 39,500 mM^−1^ cm^−1^ at 260 nm, respectively. The construct was mixed with the linearized plasmid (see above) at molar concentrations of 2 nM and 0.5 nM, respectively, in 20 mM Tris-HCl (pH 7.5), 100 mM NaCl, 10 mM MgCl_2_, 10 mM DTT, 0.5 mM ATP, and 0.02 U/µL of T4-DNA Ligase (Thermo Scientific Inc., Waltham, MA, USA). The enzymatic ligation was conducted for 20 h at 4 °C. The ligated plasmid containing the insert was separated from the enzyme and ATP on a 1 mL HiTrap Q HP column (Cytiva, USA). The incubation was loaded onto the column at a flow rate of 0.5 mL/min. The column was subsequently washed with 10 mL of 0.1M KCL and 10 mM HEPES-K (pH 7.5). The plasmid, pUC(5A/5A), was eluted from the column in 1 mL of 1.5 M KCL and 10 mM HEPES-K (pH 7.5). The eluted plasmid was chromatographed on a desalting NAP-5 column (Cytiva Inc., Marlborough, MA, USA) equilibrated with 50 mM KCl, 10 mM HEPES-K (pH 7.5).

pUC(5A/5A) was conjugated with ATP-NPs (or other particles tested) as follows. The eluted plasmid was mixed with 1000-fold molar excess of ATP-NPs in 50 mM KCl and 10 mM HEPES-K (pH 7.5), and the mixture was incubated for 16 h at 25 °C. Incubation of the plasmid with 5 nm particles was carried out for 16 h at 25 °C in 10 mM citrate-K (pH 6). The conjugates were separated from the excess of particles, on a size-exclusion Sepharose CL-2B (GE HealthCare Ltd., Chicago, IL, USA) (5 × 0.9 cm) equilibrated with 10 mM KCl and 5 mM HEPES-K (pH 7.5); 1 mM BSPP was added just before the loading of the incubation containing 5 nm GNPs onto the column. The void volume fraction containing the conjugate was collected and scanned by AFM.

### 2.7. Electrophoresis

Samples were loaded onto 4% (unless mentioned otherwise) agarose gel (7 × 7 cm^2^) and electrophoresed for 1 h at 4 °C and 100 V. Tris−acetate−ethylenediaminetetraacetic acid (TAE) in addition to being used to prepare the agarose also served as the running buffer. 

### 2.8. Atomic Force Microscopy (AFM) 

ATP-NPs (absorbance at 420 nm ≈ 5 mAU) or pUC19-NP conjugates (absorbance at 260 nm = 5 mAU) in 1 mM Mg-Ac were poured on a freshly cleaved mica and deposited for 5 min under ambient conditions. The surface was subsequently rinsed with 1 mL of ice-cold DDW and quickly dried by a stream of nitrogen gas. AFM imaging was performed on a Solver PRO AFM system (NTEGRA SPECTRA II, NT-MDT Ltd., Moscow, Russia) in a semi-contact (tapping) mode using High Accuracy Non-Contact AFM probes from the PHA-NC series (ScanSens, Berlin, Germany). The images were “flattened” (each line of the image was fitted to a second-order polynomial, and the polynomial was then subtracted from the image line) with Nova image processing software (NT-MDT Ltd. Moscow, Russia). The images were analyzed using imaging software programs: WSxM Nanotec Electronica S.L (WSxM v4.0 Beta 10.0, Nanotec Electronica Ltd., Madrid, Spain) [36] and SPIP software (MountainsSPIP^®^8, Image Metrology A/S, Hørsholm Denmark). 

### 2.9. HR-TEM of ATP-NPs

HR-TEM images were obtained with a Thermo Fisher Scientific Talos F200i transmission electron microscope instrument. The sample was prepared by dropping 4 µL of ATP-NPs solution (absorbance at 420 nm = 2.7 AU) on an ultrathin (3–4 nm) carbon-coated copper grid. After 1 min, the solution was pulled out by touching the edge of the grid with filter paper.

## 3. Results and Discussion

### 3.1. Synthesis and Characterization of ATP-NPs

We showed by high-resolution TEM (HR-TEM) and AFM analysis that the reduction of gold ions by borohydride in the presence of ATP led to the formation of very small (about 2 nm in diameter) uniform gold nanoparticles, ATP-NPs (Figure 1), if the following criteria were satisfied: 1, the concentration of gold ions was lower than 20 µM, and 2, synthesis was conducted at mild alkaline pH (8–9). Synthesis, if conducted at neutral or mild acidic pH, yielded non-uniform particles varying in diameter from 2.5 to 4.5 nm. The average diameter of ATP-NPs synthesized under former conditions (low ATP concentration, alkaline pH) was equal to 1.5 ± 0.5 nm or 2 ± 0.4 nm as estimated by AFM or HR-TEM image analysis (Figure 1A,B, respectively). The slightly lower value measured by AFM was most likely due to the interaction of the deposited objects with mica and the AFM cantilever, causing compression deformation of the particles. This deformation was previously shown to strongly affect the measured height of soft biomolecules [37,38]. The particles showed no noticeable absorption peak at ~500 nm (Figure 2A), which was attributed to localized surface plasmon resonance in gold nanoparticles. The intensity of the peak was strongly reduced with the decrease in the particle’s size and can be clearly noticed in the absorption spectrum of GNPs bigger than 2 nm in diameter [25,39,40,41,42,43,44]. The lack of this peak was in line with the small size of ATP-NPs measured by AFM and HR-TEM. 

To estimate the extinction coefficients of the particle at specific wavelengths and to determine the number of gold atoms composing it, we employed the following approach. The particles were conjugated to a 38-base single-stranded (ss)DNA thiolated at the 5′ end. Incubation with 2.5 molar excess of the strand yielded ATP-NP–oligonucleotide conjugates composed of different numbers of strands. The individual conjugates bearing discrete numbers the oligonucleotide strands were separated from one another by agarose gel electrophoresis (Appendix A). As expected, the mobility of conjugates decreased with the increased number of strands attached to the particle (Appendix A). The individual conjugates were electroeluted from the gel, and their absorption spectra were analyzed (Appendix A). Each individual conjugate absorption (at any wavelength) was equal to the sum of the particle’s absorption and that of the DNA strand multiplied by the number of strands in the conjugate. Hence, by subtracting the absorption of the conjugate bearing one strand per particle (blue curve in Appendix A) from the spectra of the conjugates bearing two strands (black curve in Appendix A), one can obtain the spectrum of a sole strand in the conjugate. This spectrum (black continuous curve in Appendix A) corresponds nicely to the spectrum of 0.31 µM strands (dashed curve in Appendix A). We thus conclude that the concentration of conjugates (bearing either one or two strands; blue and black curves in Appendix A, respectively) is equal 0.31 µM too. The extinction coefficients of the particle in the visible range of the spectrum (DNA does not contribute to the absorption spectrum at wavelengths longer than 320 nm) can then be calculated.

For instance, the calculated extinction coefficient of the particle at 420 nm (OD is equal to 0.23 AU; see Appendix A) is equal to $\varepsilon_{420\;\mathrm{nm}}=$7.42 × 10^5^ M^−1^cm^−1^ (AU/[ATP-NPs]; 0.23/0.31 × 10^−6^ M). This value is about five times higher than that (1.56 × 10^5^ M^−1^cm^−1^) known for a 1.4 nm (in diameter) Au_55_ particle used widely in research. This is in good agreement with the greater (~2 nm in diameter; Figure 1) size of the former NPs.

To estimate the number of gold atoms composing ATP-NP as well as the number of ATP residues associated with it, we dissolved the particles with CN^−^ and analyzed the spectrum of the CN-treated sample. As seen in Appendix A, incubation of ATP-NPs (black curve) with 10 mM KCN for 5 min led to complete bleaching of the particles and the appearance of two sharp peaks with maxima at 240 and 230 nm, corresponding to Au(CN)_2_^−^ [46,47,48] (red curve). A peak at about 260 nm corresponding to ATP became clearly noticeable (red curve in Appendix A). The spectrum of 0.43 µM CN-treated particles (Figure 3B, red curve) was successfully simulated (Figure 3B, black dashed curve) as a sum of 107.7 μM Au(CN)_2_^−^ and 16.9 μM ATP spectra (Appendix A, gray curves). The molar concentrations of ATP-NPs, Au(CN)_2_^−^, and ATP were estimated here using the following extinction coefficients: 7.4 × 10^5^ M^−1^ at 420 nm (Appendix A), 3.9 × 10^3^ M^−1^ at 240 nm [46], and 15.4 × 10^3^ M^−1^ at 260 nm [49], respectively. To conclude, the above analysis showed that ATP-NP was composed of 250 gold atoms and 39 ATP molecules. Assuming that the particles are spherical, with face-centered cubic packing of the gold atoms, and that their structure and lattice number (nfcc) are equal to 59 Au nm^−3^, the gold core diameter (*D*) and the number of atoms in the particle (*N*) can be linked by Equation (1) [40].
(1)$$D={\left(\frac{6N}{{\pi n}^{fcc}}\right)}^{1/3}={\left(\frac{6N}{{\pi \times 59}_{{\mathrm{nm}}^{-3}}}\right)}^{1/3}$$

The diameter of a gold nanoparticle comprising about 250 gold atoms calculated using Equation (1) should then be equal to about 2 nm, which agrees with the estimated value for ATP-NP using HR-TEM (Figure 1). The similarity between the absorption spectra of ATP-NPs (Appendix A, black curve) and that of the 2 nm GNPs protected by thiol-containing and phosphine ligands [40,41,42], as well as the diameter of about 1.9 nm estimated for the thiol-capped NP composed of 200 gold atoms [40,50], further confirms the estimated diameter of ATP-NP.

The ATP residues coating the particle were tightly bound to the core and could not be removed from the particle’s surface either by size-exclusion chromatography or by ultrafiltration. Interestingly, the dissolving of ATP-NP with CN^−^ led to the appearance of a clear sharp peak at around 260 nm (Appendix A, red curve; notice the absorption changes in the 240–280 nm region of the spectrum), which was attributed to ATP, while the spectrum of ATP conjugated to the particle was not as pronounced (Appendix A, black curve). This observation emphasizes the strong effect of the nanoparticle’s core on the absorption of light by ATP directly and tightly bound to the gold surface. In the scope of this work, we will not investigate this phenomenon in depth. 

ATP-NPs are characterized by very high stability under various conditions and do not aggregate at salt concentrations of up to 1M. They can be lyophilized and stored in a dry form for months at 4 °C (Appendix A). The capping ligand (ATP) in the particles can be easily and rapidly substituted by stronger ligands such as phosphines, thiol-containing compounds, and DNA oligonucleotides (Appendix A). The particle core is also easily accessible from the surrounding medium. This leads, in particular, to fast dissolution of the particle by CN^−^. As seen in Appendix A, the complete dissolution of ATP-NPs by 10 mM KCN took place within 1 min (Appendix A, red and blue curves), while almost no dissolution of commercial Au_55_ nanoparticles, coated with thiolated ligands, was seen even after half an hour treatment with CN^−^ (Appendix A, black and green curves). Incubation with Bis(p-sulfonatophenyl)phenylphosphine (BSPP) considerably reduced the reactivity of ATP-NPs; BSPP-treated particles were resistant to CN^−^ and were dissolved by cyanide only after 2 days of incubation under ambient conditions. 

### 3.2. Enlargement of ATP-NPs

The high reactivity of ATP-NPs made it possible to enlarge ATP-NPs by surface-catalyzed reduction of gold ions on the surface of the particle. Binding of newly reduced gold atoms to the particle surface was strongly preferred over the formation of new particles, leading to ATP-NP growth. We demonstrated (Figure 2) that 1-min incubation of the particles with a mixture of gold ions and ascorbate resulted in bigger uniform particles. The final particle size could be tuned by altering the concentration of gold ions in the enlargement mixture. As seen in Figure 2, the enlarged particles moved as single narrow bands in the gel. 

The color of the bands changed from brown (typical to the particles of about 1.5–2 nm in diameter) to wine-red as the particle size increased (Figure 2A, lanes 2–4). Figure 2D shows the absorption spectra of the particles electroeluted from colored areas in the gel. The intensity of the absorption peak at ~500 nm grew with the particle size. This approach (Figure 2) enabled the production of highly uniform particles of any desired size in the 2–4 nm range. Enlargement of commercial Au_55_ particles (strongly coated with thiolated ligands comprising carboxylic acid groups) under identical conditions led to formation of non-uniform particles with a wide size distribution (Appendix A).

### 3.3. ATP-NPs Binding to DNA 

The advantage of ATP over strong capping ligands (most commonly phosphines and thiols) is that the nucleotide triphosphate can be rather rapidly (hours) replaced by various ligands yielding small GNPs conjugated to molecules of interest. In particular, ATP-NPs, being highly reactive, can bind specifically to single-stranded (ss) tag sequences at the ends of dsDNA molecules, as well as to mismatched regions in the double-helix DNA (Figure 3 and Figure 4). In the experiments illustrated in Figure 3 and Figure 4, pUC19 was linearized by two restriction enzymes, EcoRI and HindIII. The linearized ds plasmid, comprising short (four bases) overhangs at the 5′ end of each strand, was incubated with 1000-fold molar excess of ATP-NPs. DNA–nanoparticle conjugates were separated from excess ATP-NPs by size-exclusion chromatography as described in Materials and methods and imaged by AFM. Tiny bright dots corresponding to ATP-NPs can be seen (though hardly) at the ends of DNA molecules (Figure 3A). The small size, similar to that of the double-helix diameter, did not allow reliable detection of the conjugated particles by conventional AFM imaging techniques. The particles, however, became clearly visible after the treatment of the conjugate with gold ions and ascorbate (Figure 3B). To test the ability of ATP-NPs to interact with DNA regions characterized by local structural alterations of the helix, we inserted a short (35 base pairs) ds oligonucleotide containing a mismatched fragment (5A/5A) into pUC19 (see Appendix A for illustration). It has been demonstrated earlier that GNPs protected by citrate efficiently bind to ssDNA and do not bind to dsDNA [51,52]. We demonstrated that the incubation of pUC19(5A/5A) with ATP-NPs (Figure 4B and Appendix A) resulted in the binding of a single particle per plasmid. As in the case of the linearized plasmid (Figure 3A), small bright dots, corresponding to the DNA-bound nanoparticles, can be noticed in some of the plasmids (Figure 4A). The particles became clearly visible (bright dots associated with DNA) after their enlargement with gold ions and ascorbate (Figure 4B). We also showed that the incubation of the plasmid lacking a 5A/5A sequence with ATP-NPs for 16 h under identical conditions did not result in formation of the conjugate (Appendix A). These results, all together, were consistent with specific binding of ATP-NPs to the non-hybridized 5A/5A region in the plasmid. We also demonstrated that despite their small size (1.4 nm in diameter), commercial Au_55_-NPs coated with thiolated ligands were incapable of binding to pUC19(5A/5A). 

Incubation of these particles with pUC19(5A/5A) did not lead to formation of the DNA–NP conjugate (Appendix A). The same was true for ATP-NPs pre-treated with BSPP (Appendix A) and for bigger (5 nm) citrate-protected GNPs (Appendix A). 

## 4. Conclusions

Based on the above results we conclude that in order to specifically bind to partially hybridized regions in DNA characterized by alterations of the helix, the particle should meet the following criteria: 1, it should be less than 2 nm in diameter, and 2, it should not be protected by strong ligands (i.e., thiols or phosphines) that cannot be readily exchanged by the nucleic bases. The ATP-NPs reported here are therefore ideally suited for the purpose of detection of these regions in dsDNA. The non-canonical regions, resulting from insertions and deletions, as well as regions comprising point mutations, are commonly present in cancer cells, leading to genomic instability and chromosomal abnormalities [53,54,55,56]. Moreover, selective binding of ATP-NPs to non-canonical regions in the genome might possibly suppress DNA repair and as a result selectively eradicate cancer cells and tumors. To reduce viability, the particles should be specifically delivered into the pathological cells. This can be done, in particular, by encapsulation of the particles in receptor-targeted liposomes, followed by delivery of the cargo vesicles into the cell, an experimental strategy that we have used before for delivering various bio-active compounds into cancer cells [57,58]. ATP-NPs are small enough to enter the cell nucleus from the cytoplasm by passive diffusion. Binding of ATP-NPs to DNA areas comprising mutations might obstruct or even completely block the nucleic acid repair mechanism causing cell death. The particles can also be useful in the mapping of mismatched regions in dsDNA. Incubation of dsDNA extracted from healthy or cancerous cells followed by AFM imaging of the conjugates as described in Figure 4 can enable highly sensitive detection of non-canonical regions in both types of cells. Using this strategy, one might be able to rather rapidly differentiate between healthy and pathological cells.

## Figures and Tables

**Figure 1 nanomaterials-13-03080-f001:**
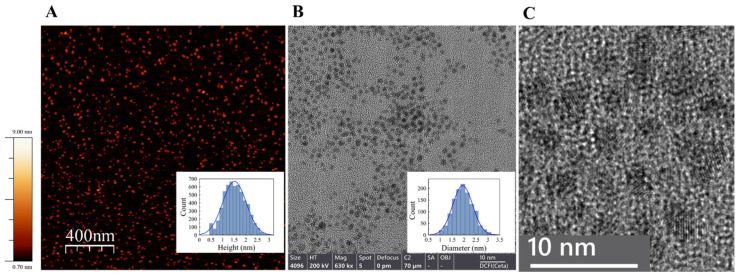
AFM (**A**) and HR-TEM (**B**,**C**) characterization of ATP-NPs. (**A**) The particles were deposited on a freshly cleaved mica and scanned as described in Materials and methods. The inset presents a height analysis of the particles by Scanning Probe Image Processor Software (SPIP, MountainsSPIP^®^8) and the Gaussian fit. The estimated average height (diameter) of the particles (5699 particles were analyzed) was equal to 1.5 ± 0.5 nm. (**B**) HR-TEM measurements were conducted using Talos F200i TEM. The particles were deposited on a thin carbon copper grid and imaged as described in Materials and methods. The inset presents statistics of particle diameter distribution measured by ImageJ software (software version 1.53) and the Gaussian fit. The estimated average diameter was equal to 2 ± 0.4 nm (1217 particles were analyzed). (**C**) Ultrahigh-resolution TEM scan of the particles. A gold lattice is seen in the individual particles. Using ImageJ software, the lattice spacing distance of 0.235 nm was estimated. This value was consistent with the value of 0.23–0.25 nm estimated for the gold face-centered cubic (fcc) packing crystal structure. Similar values were also estimated for small thiol-protected GNPs [28,45].

**Figure 2 nanomaterials-13-03080-f002:**
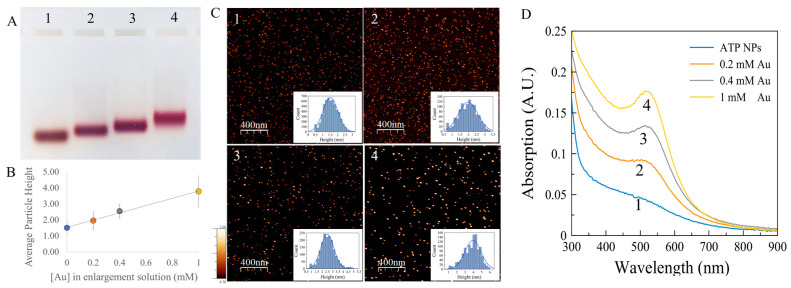
ATP-NP growth in the presence of Au ions and ascorbate. (**A**) Electrophoresis of the enlarged particles. ATP-NPs (lane 1) were synthesized as described in Materials and methods; 200 nM of ATP-NPs were incubated with 1 mM ascorbate and 0.2 (lane 2), 0.4 (lane 3), or 1 mM (lane 4) HAuCl_4_ (see Materials and methods for details). The particles were concentrated by ultrafiltration and electrophoresed on 4% agarose gel at 100 V in TAE buffer at 15–20 °C. The particles electroeluted from colored areas in the gel were imaged by AFM as described in Materials and methods. Panels 1, 2, 3, and 4 in C correspond to lanes 1, 2, 3, and 4 in panel A, respectively. The insets (in **C**) show the particles’ height distribution. The average heights (diameter) of the particles electroeluted from lanes 1, 2, 3, and 4 were equal to 1.5 ± 0.5 (*n* = 5699), 2 ± 0.5 (*n* = 1134), 2.5 ± 0.4 (*n* = 1744), and 3.8 ± 1 nm (*n* = 1154), respectively (n is a number of measured nanoparticles). (**B**) The dependence of the average diameters of the particles on the concentration of gold ions in the enlargement assay. (**D**) Absorption spectra (curves 1, 2, 3, and 4) of the particles electroeluted from lanes 1, 2, 3, and 4, respectively.

**Figure 3 nanomaterials-13-03080-f003:**
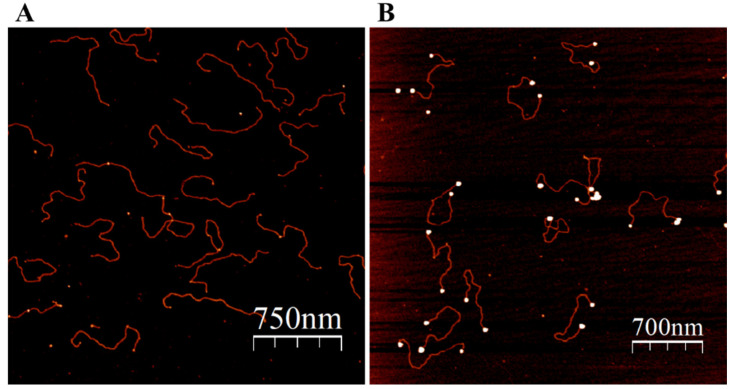
Binding of ATP-NPs to the ends of pUC19. The plasmid was cleaved by EcoRI and HindIII according to standard protocol. A 1000-times molar excess of ATP-NPs was added to the plasmid and incubated for 16 h in 50 mM KCl and 10 mM HEPES-K (pH 7.5) under ambient conditions. The incubation was chromatographed on a Sepharose CL-2B column (see Materials and methods). (**A**) The void volume fraction (OD at 260 nm is equal to 0.1–0.2 AU) containing the plasmid was diluted 20 times with 10 mM Mg-Acetate; 50 µL of the diluted solution was deposited on a freshly cleaved mica for 5 min, subsequently rinsed with double-ionized water (DDW), dried with N_2_ gas, and scanned by AFM. (**B**) The deposited molecules (as in “A”) were enlarged as described in Materials and methods (“Enlargement of ATP-NP on mica surface”), rinsed with double-ionized water (DDW), dried with N_2_ gas, and scanned by AFM.

**Figure 4 nanomaterials-13-03080-f004:**
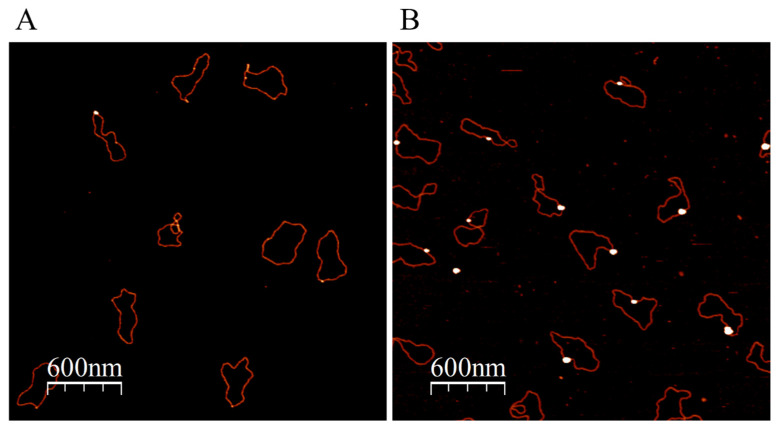
Binding of ATP-NPs to the 5A/5A mismatched region inserted into Puc19. The plasmid, pUC(5A/5A), was prepared as illustrated in Appendix A and described in Materials and methods. pUC(5A/5A) was conjugated with ATP-NPs and separated from the excess of particles as described in Figure 3. Deposition of the conjugate and enlargement of the particle associated with the plasmid was also conducted as in Figure 3. AFM images of the conjugate before (**A**) and after (**B**) enlargement of the particles.

## Data Availability

Data sharing is not applicable to this article.

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
