# Peer review of "Ultrasmall ATP-Coated Gold Nanoparticles Specifically Bind to Non-Hybridized Regions in DNA"

_nanomaterials, 2023, doi:10.3390/nano13243080_

Round 1

Reviewer 1 Report

Comments and Suggestions for Authors

This manuscript reported a method for synthesis ultra-small, highly uniform ATP-coated gold nanoparticles, ATP-NPs. The analytical content of this manuscript is interesting. However, it is better to have further adjustments and supplements.

1.        The manuscript has only four sections, so the chapter title of "Section 5" needs to be adjusted. 

2.        There are some other related literatures about the combination of GNPs and DNA oligonucleotides, such as "DOI10.1039/c6en00323k", " DOI10.1007/s00604-018-3028-7"," DOI10.1021/acs.analchem.9b02145".

3.        Sections 2.8 and 2.9 should be placed in the Supplementary Materials.

4.        The structure of the third part of the manuscript is not clear, and the subtitle should be added.

5.        The figures of the full manuscript should be readjusted: (1) the style of the icon ruler in Figure 1 and Figure 3 has not been unified; (2) the layout of Figure 2 is not enough to clearly display all the contents, and the clarity is not enough; (3) figures title should be as concise as possible.

6.        The manuscript lacks materials to support the stability and selectivity of ATP-NPs.

7.        The manuscript does not provide the application of ATP-NPs in real samples. Therefore, it is necessary to provide relevant experiments.

Comments on the Quality of English Language

Minor editing of English language required

Author Response

Reviewer 1

Comments and Suggestions for Authors

This manuscript reported a method for synthesis ultra-small, highly uniform ATP-coated gold nanoparticles, ATP-NPs. The analytical content of this manuscript is interesting. However, it is better to have further adjustments and supplements.

We thank the reviewer for his positive assessment.  

  1. The manuscript has only four sections, so the chapter title of "Section 5" needs to be adjusted.

We thank the reviewer for noticing this mistake! Section 5 is changed to Section 4 (row 414) in the amended version of the manuscript.

  1. There are some other related literatures about the combination of GNPs and DNA oligonucleotides, such as "DOI10.1039/c6en00323k", "DOI10.1007/s00604-018-3028-7"," DOI10.1021/acs.analchem.9b02145".

Yes, there are thousands of papers on the interaction of DNA oligonucleotides and GNPs. Actually, this topic is one of the most popular in nanotechnology in the last 3 decades. Motivated by the discoveries of Chad Mirkin and Paul Alivisatos (two papers were published by these researchers back in 1996 in Nature). Many research laboratories moved to this fascinating field.  There are many papers devoted to the interaction of either short DNA oligonucleotides with GNPs (mostly ranging in diameter from 5 to 50 nm) and much less on the interaction of the particles with the double-stranded DNA. The reason for this is pretty simple, no such interaction is taking place and bare GNPs can interact exclusively with single-stranded DNA oligonucleotides. The unique advantage of the particles reported here is that they are capable of interaction with very small (composed of a small number of nucleotides) perturbed areas in the double helix. Neither bigger (5-50 nm in diameter) bare GNPs nor small (1.4-3 nm in diameter) strong ligand-coated GNPs are capable of interaction with the double helix. No literature is available (to the best of our knowledge) on GNP-specific interaction with mismatched areas in the double helix. As for the papers recommended for citation by the reviewer, we don’t see much of their relevance to the results presented in the manuscript. We would be very grateful for clarification on this matter.   

  1. Sections 2.8 and 2.9 should be placed in the Supplementary Materials.

We don’t agree with the reviewer’s opinion/suggestion. In the section’s technical issues of AFM and TEM measurements are address, so there is no any rationale for moving the sections to the SM. This could be ok moving the whole section (Materials and Methods) to the SM. But this is usually done for articles but not for regular manuscripts. 

  1. The structure of the third part of the manuscript is not clear, and the subtitle should be added.

We are thankful to the reviewer for this comment. Subtitles have been added (rows: 224, 265 and 331 of the amended version of the manuscript). The paper is, indeed, better organized now.  We appreciate the reviewer once again.

  1. The figures of the full manuscript should be readjusted: (1) the style of the icon ruler in Figure 1 and Figure 3 has not been unified.

Yes, indeed the scale bars on either Fig. 1 or Fig. 3 are not unified. As for Fig. 1, they in principle couldn’t be! Panel “C” shows high-resolution TEM image of the particles in which even individual atoms of the particle’s lattice can be seen. Panel “A” is actually presenting a blown-up image of an area from panel “B”. It’s clear that initial and blown-up images can’t have the same scale. As for Fig. 3, the aim of the figure is to show the binding of the particles (bright spots in the image) to the DNA ends rather than comparing the contour lengths of DNA and DNA-NP molecules. In this case, there is no need to unify the scale bars. Moreover, the scale bars are not much different (700 and 750 nm). I hope that we got the point right and this is not a difference in the size of the letters (nm) in the panels. If yes, this point might be addressed by the production team, but we believe that this will not be the case.    

 (2) the layout of Figure 2 is not enough to clearly display all the contents, and the clarity is not enough

Corrected in accordance with the reviewer’s comment. We appreciate the reviewer’s suggestion. 

 (3) figures title should be as concise as possible.

Corrected in accordance with the reviewer’s comment. We believe that it is now (“Binding of ATP-NPs to the ends of pUC19) is “as concise as possible”. We appreciate the reviewer’s suggestion. 

  1. The manuscript lacks materials to support the stability and selectivity of ATP-NPs.

That’s not correct, see rows 317-319: “ATP-NPs are characterized by very high stability under various conditions and do not aggregate at salt concentrations of up to 1M. They can be lyophilized and stored in a dry form for months at 4°C (Figure S3)” for stability and Figure S7 for selectivity.  

  1. The manuscript does not provide the application of ATP-NPs in real samples. Therefore, it is necessary to provide relevant experiments.

The reviewer is right, the work is not aimed at demonstration of the particle's effect on cells and animals. However, demonstration of the effect of the particles on living systems is a goal of future research to be perused in the laboratory. At present, we have reported novel particles with a unique ability to bind to mismatched areas in the DNA double helix and propose their possible preferential binding to DNA of pathological cells containing many more mismatched areas compared to healthy cells. We believe that the results presented in the manuscript are novel, interesting, and important enough, to be published in “Nanomaterials”.

Reviewer 2 Report

Comments and Suggestions for Authors

General comments

This paper presents a compelling synthesis process, complemented by a meticulous characterization of particles with a substantial particle count. However, the overall process leading to the statement, 'Using this strategy, one might be able to rather rapidly differentiate between healthy and pathological cells,' is not explicitly elucidated. Providing a clearer exposition of the global process would enhance the paper's coherence and strengthen its impact, particularly by highlighting the potential for swift differentiation between healthy and pathological cells as a notable outcome.

Specific comments

Lines 39-40: “making them resistant to high (including physiological) salt concentrations.” Citation needed or be less radical in this statement. They can be more resistant to higher concentrations, but there are limits.

Lines 92-95: Is the suggested detection best executed through AFM, mirroring the paper's analysis? Or could simpler methods like UV-Vis or Fluorescence (spectral changes) offer viable alternatives? Please briefly discuss this.

Lines 246 and 250: Is there a specific reason for presenting the sizes as 1.51 ± 0.47 nm and 1.96 ± 0.41 nm and not 1.5 ± 0.5 nm and 2.0 ± 0.4 nm, respectively? See also other cases (e.g., lines 346-347).

Line 249: “ImageJ software” – please add the version of the software.

Comments on the Quality of English Language

Minor fixes by the editorial team should be enough.

Author Response

Reviewer 2

General comments

This paper presents a compelling synthesis process, complemented by a meticulous characterization of particles with a substantial particle count. However, the overall process leading to the statement, 'Using this strategy, one might be able to rather rapidly differentiate between healthy and pathological cells,' is not explicitly elucidated. Providing a clearer exposition of the global process would enhance the paper's coherence and strengthen its impact, particularly by highlighting the potential for swift differentiation between healthy and pathological cells as a notable outcome.

We are grateful to the reviewer for his comments, suggestions, and corrections. 

The reviewer is right, that the impact of the study on the field of nanobiology would be greater if will included the data on the specific eradication of pathological cells by the particles. However, demonstration of the effect of the particles on living systems is a goal of future research to be perused in the laboratory. At present, we have reported novel particles with the unique ability to bind to mismatched areas in the DNA double helix and propose their possible preferential binding to DNA of pathological cells containing many more mismatched areas compared to healthy cells. We believe that the results presented in the manuscript are novel, interesting, and important enough, though to be published in “Nanomaterials”.

Specific comments

Lines 39-40: “making them resistant to high (including physiological) salt concentrations.” Citation needed or be less radical in this statement. They can be more resistant to higher concentrations, but there are limits.

We appreciate the reviewer for noticing this. Corresponding changes (inc. reference) have been made; see lines 39-40 and ref. 17 of the revised version.

Lines 92-95: Is the suggested detection best executed through AFM, mirroring the paper's analysis? Or could simpler methods like UV-Vis or Fluorescence (spectral changes) offer viable alternatives? Please briefly discuss this.

The binding of a single nanoparticle (especially 2-3 nm one) to a long DNA molecule can’t be registered by canonical absorption spectroscopy. Fluorescence is also not applicable here, since our particles are not fluorescent. AFM is a very sensitive and powerful technique allowing to monitoring of single molecule events. Here we have very clearly shown by AFM that ATP-NP is specifically bound to a mismatched region introduced into Puc19. The spectroscopic methods (even very sophisticated) wouldn’t be able to show this!  

Lines 246 and 250: Is there a specific reason for presenting the sizes as 1.51 ± 0.47 nm and 1.96 ± 0.41 nm and not 1.5 ± 0.5 nm and 2.0 ± 0.4 nm, respectively? See also other cases (e.g., lines 346-347).

No, there was no any specific reason, so corresponding changes have been (see rows: 231, 246, 251, and 346 of the amended manuscript).

Line 249: “ImageJ software” – please add the version of the software

The version is added (see row 250 of the amended manuscript)